# Long-Term Vegetation Changes and Socioeconomic Effects of River Engineering in Industrialized Areas (Southern Poland)

**DOI:** 10.3390/ijerph20032255

**Published:** 2023-01-27

**Authors:** Oimahmad Rahmonov, Weronika Dragan, Jerzy Cabała, Robert Krzysztofik

**Affiliations:** 1Institute of Earth Sciences, Faculty of Natural Sciences, University of Silesia, Będzińksa 60, 41-200 Sosnowiec, Poland; 2Institute of Social and Economic Geography and Spatial Management, Faculty of Natural Sciences, University of Silesia, Będzińska 60, 41-200 Sosnowiec, Poland

**Keywords:** mining operation, ecosystem degradation, river engineering, urban river, post-industrial sites

## Abstract

The exploitation of mineral resources associated with human mining activities leads to the degradation of both terrestrial and aquatic biocenotic systems. The drastic disturbance of water relations as a result of the relocation of the riverbed of the Biala Przemsza River (southern Poland) for coal and filler sand mining will lead to changes in plant ecosystems. The purpose of this study was to determine and compare the diversity and distribution of vegetation in the Biała Przemsza valley in sections of channel straightening with the old riverbed and areas undisturbed by engineering works against the background of land use in temporal and spatial aspects. The results of the ecological and phytosociological studies showed that the composition of flora and vegetation types varied. Within the transformed riverbed, anthropogenic mixed forests with species characteristic of different ecological systems are developing, whereas the non-regulated section of the river is overgrown by an alder riparian forest with an almost complete species composition for this plant community. The highest Simpson’s biodiversity index was found in the anthropogenically disturbed section of the river (0.86), and in the undisturbed section, it was 0.83. Both sections of the river were dominated by species of the family *Compositae*, *Poaceae*, *Caryophyllaceae*, *Rosaceae* and *Apiaceae*. The diversity of the flora in the transformed sections of the valley is determined by the presence of mosaics and microhabitats, as well as the nature of the surrounding vegetation, which is reflected in the ecological requirements of the flora concerning light preference (moderate light [56.25%]), and almost 90% of the flora from the area of the regulated section of the valley develops on humus-poor and mineral-humus soils. Although this area has lost its original natural function, it is now valuable for selected economic and social functions, especially in highly urbanized regions.

## 1. Introduction

The natural environment and its functioning increasingly depend on human activity [1,2,3,4] and its intensity. Anthropogenic pressure should be considered with respect to both the sustainable use of natural resources and the undertaking of various kinds of protective measures [5]. Human economic activity directed towards the inexorable extraction of mineral resources has led and continues to lead to the disintegration of aquatic and terrestrial ecosystems. As a result, individual elements of the landscape are subject to far-reaching transformations and often lose their ecosystem services [6]. One of the effects of human economic activity is the creation of completely new habitats, such as post-industrial areas. These areas are formed on large spatial and temporal scales [7]. Today, they occupy about 6% of the earth’s surface [8]. It is estimated that mining activities (hard coal, metal ores, aggregates) have affected about 1% of the land [9], so post-mining areas are an important component of the landscape of many places around the world. For some cities and regions, this type of area is fundamental [10,11]. New anthropogenic forms are unique habitats, because they are a very poor mineral substrate, often devoid of organic matter [12]. Among the most common anthropogenic forms in southern Poland are open-pit sand mines.

Open-cast mining, which results in the complete disintegration of biocenoses and the pedosphere, is one of the human activities that has the most drastic impact on the environment. It involves the removal of vegetation and the destruction of soil cover in areas from which minerals, including sand, are extracted. As a raw material, sand is one of the most widely used mineral resources on Earth [13]. Sand is an important mineral for various industries around the world and a habitat for many organisms [14]. Sand mining is becoming an environmental issue, as there is an increasing demand for sand for industry, construction and other purposes [15,16,17,18] and as a backfill material to fill voids in post-mining corridors (underground mining) to prevent subsidence [19].

In order to protect the ground surface from deformation, the development of coal mining in urbanized areas in southern Poland has required the use of sand filling in underground galleries. The Pleistocene sands overlying Upper Carboniferous deposits were a natural resource base for hydraulic backfill. The increase in demand for backfill sands resulted in the expansion of sand mining areas in many areas of the Upper Silesian Coal Basin (USCB). Riverbeds, communication roads and various infrastructure elements were used in the process. A similar problem affected zinc and lead ore mining in southern Poland and coal mining in Germany [20].

Nowadays, due to the increasing demand for mineral resources, the number of degraded and disturbed areas has increased worldwide [21,22,23,24,25], and open-pit mining and related works often have an irreversible impact on nature. Mineral mining has one of the strongest impacts of all industries, distorting the environment globally [26,27] and often causing the loss of its ecological functions [28,29]. One of these negative impacts is the alteration of water relations in the zones around sand mining activities. Mineral extraction often requires the drainage of the surrounding area, which has a hugely negative impact on wetland ecosystems.

Drainage of river valleys, including river channel beds, is a transformative process for wetland ecosystems [30]. This anthropogenic interference with the natural environment has taken place and continues to take place over large areas in many regions of the world, in the valleys of both medium-sized and small rivers [31,32,33,34]. Many publications have been devoted to the natural effects of river straightening and related ecosystems [5,35,36,37] and their further functioning and restoration [38,39,40,41].

Most hydro-technical engineering works, including the straightening of the riverbed by concreting, the artificial construction of dikes, land reclamation and drainage, have caused the vegetation of riparian wetland ecosystems to be completely transformed and replaced by other communities [42,43]. In addition, significant amounts of municipal and industrial wastewater have been discharged into rivers, causing pollution. The consequence of the degradation of the ecosystems has been the reduction or even complete loss of their ecosystem services [27,28,34]. These changes are more evident in vegetation communities and their species composition. Vegetation is one of the most sensitive components of the environment and can be used as an indicator of changes in both the biotic and the abiotic conditions of the river ecosystem.

Anthropogenic disturbance of riparian ecosystems leads to a reduction in biodiversity and synanthropization of the existing vegetation [35,42]. This often manifests as an increase in the number of anthropogenic communities, the disappearance of primary species combinations, the creation of new heterogeneous species combinations, the formation of species-poor and less diverse communities and even artificial tree planting through forest restoration.

Floodplain vegetation provides many ecosystem services, such as protection against soil erosion, water retention, transport of matter, self-purification of rivers and habitats for many organisms. In the industrial areas of the city, areas of floodplain vegetation are almost the only facilities that serve as a place for recreation and tourism and have an undeniably high aesthetic value [44,45]. The transformation of river valleys as a result of water engineering can cause them to completely lose their ecosystem and social functions [5,28].

In the present study, it was hypothesized that the development of vegetation and its species composition in the new areas created by river channel relocation would differ from the natural river sections. It was also assumed that the long-term transformation of the natural environment around water features has an impact on socio-economic and recreational aspects in urban areas as well. Hence, the objective of this study was to determine and compare the diversity and distribution of the present vegetation in the Biała Przemsza valley consisting of anthropogenic (straightening the riverbed) and natural sections against the changing background of land use in the region.

## 2. Materials and Methods

### 2.1. Study Area

The study area is located in southern Poland in the NE part of the USCB (Upper Silesian Coal Basin). The phytosociological study area is located in the southern part of the Maczki-Bór sand deposit (Figure 1). The left bank of the river has been exploited since 1920 in the sand pit “Jęzor Północ”, and the sand pit “Maczki-Bór” was added on the right bank in 1952. The analyzed area includes the current and former bed of a river flowing in an area affected by coal mining and sand exploitation.

In the early 1970s, in order to widen the mining area of the Maczki-Bór deposit, as well as to protect the pit against the infiltration of water from the Biała Przemsza River, the original riverbed was relocated. At the same time, the new riverbed was sealed with concrete slabs. In 1980–1990, as a result of the exploitation of increasingly deep layers of sand, the riverbed of the Biała Przemsza River was several meters higher than the level of deposit exploitation and the free water table drained by the mine. The exploitation of sands from the bottom of the Maczki Bór West and East deposits made it possible to start storing post-mining waste from underground coal mines in some parts of these pits (Figure 2).

The geological settings underlying the sand deposit are Upper Carboniferous clay-sandstone formations with coal beds that were exploited until the 1980s. A protective pillar was established for the Biała Przemsza River, and no coal was exploited there. A few hundred meters south of the study area, limestone-dolomitic formations of the Middle Triassic lie on the eroded surface of the Carboniferous rocks (Figure 2). From the 16th century to the 1920s, lead and silver ores as well as zinc were mined from Middle Triassic formations by underground methods [46]. Overlying the erosional surface from Carboniferous and Triassic sediments are fluvioglacial formations of the Pleistocene. The sandy formations fill morphological depressions in the bedrock and are up to 30–50 m thick. These are easily water-permeable variably graded quartz sands with insets of gravels with a small proportion of clay fractions. Locally, in the shallow layers of the profile, there are stagnant clay formations associated with fluvial accumulation.

The stockpiled material is Carboniferous bedrock (sandstone, shale, siltstone) with a small percentage of carbonaceous matter (Figure 2). A further 10–15% is electrowinning ash. In the Maczki-Bór West field, filling/reclamation was carried out from 1977 and was completed in 2006. In the Maczki-Bór East field, reclamation has been carried out since 2003, and the remaining sand reserves should be extracted by 2025, after which the remaining part of the pit will be recultivated. The areas of the Bór-West and Bór-East pits are 358.7 ha and 199.3 ha, respectively.

### 2.2. Cartographical Analyses

The environmental changes in the studied area, the available cartographic material [47,48,49,50,51,52,53,54] and surveys from surrounding locations were used. Map analysis and interpretation was carried out using geographical information system (GIS) methodologies, and an interpretative sketch was created in the MapInfo and QGIS geospatial programs. Second, rectification was carried out, and the image was adjusted to the reference layer using control points. Topographic maps from 1993–1996 and an orthophotomap are available in georeferenced form. The results obtained from the orthophotomaps (from 2021) were verified and supplemented in the field.

All cartographic materials were digitized, and errors usually generated during this operation were detected and eliminated. Screen digitization was combined with the creation of a database of land development. Data included in each series of maps were aggregated to make them comparable (see results). As a result, vector maps were created that allowed for the comparison of land development in particular time sections to be carried out.

### 2.3. Vegetation Analyses

In order to determine the impact of the change in water relations on the development of vegetation, caused by the shift in the riverbed, three transects were established. The length of transects and their relative height was measured with a tape measure. The first transect (Transect I) and the second transect (Transect II) start from the old riverbed floodplain and end behind the artificial embankments of the riverbed (see results), and Transect III was delineated on parts of the valley whose character remained unaffected (natural) by anthropogenic impacts. Except for this fragment in the study area, the plant communities (especially forest) are anthropogenic in character and have been formed as a result of reclamation.

On selected transect plots differing in morphology and origin (natural, e.g., an old riverbed), phytosociological surveys were carried out, and the dominant plant species on each transect section were marked on a sketch (see results).

#### 2.3.1. Vegetation Community Analyses

A survey of the vegetation was conducted between May/ June and the end of September 2019–2022 and mainly involved geobotanical and ecological research. During the research, an inventory of the flora was undertaken, and phytosociological documentation was also carried out to differentiate the vegetation communities from each other. Furthermore, field studies that involved verification of the actual range of vegetation were conducted. The plant communities distinguished as a result were plotted against a previously prepared sketch. First, the study area was divided into deciduous and coniferous forests and meadows based on the orthophotomap (Figure 1). The distinguished vegetation communities were mapped onto the previously prepared sketch.

In order to identify the plant communities in the individual distinguished patches, phytosociological relevés were made by applying the Braun-Blanquet method [55] to the transects, despite the artificial (plantation) nature of the vegetation in transects one and two. This method involves drawing up floristic lists by dividing species into woody, shrubby and herbaceous species, and then determining the degree of plant species cover on the analyzed surface using a six-point scale. According to this method, to identify vegetation at the association level, the presence of a characteristic and distinctive species for a given association is sufficient. These species are associated with very specific ecological requirements.

Plots of 100–200 m^2^ were selected for phytosociological surveys, where a list of individual species was compiled along with the area occupied according to the 6-degree Braun-Blanquet scale [55]. In the scale used, the following categories were applied: degree 5—species covering 75–100% of the area; 4—species covering 50–75%; 3—species covering 25–50%; 2–25–5%, 1–5–1% and +-species is present in the area. The Braun-Blanquet scale was converted to percentages as follows: +–0.1%, 1–5%, 2–17.5%, 3–37.5%, 4–62.5%, 5–87.5%. Simpson’s floristic biodiversity index [56] was calculated within the study areas, according to the formula:$$D=\frac{\Sigma n\left(n-1\right)}{N\left(N-1\right)}$$
where n is the number of individuals of a given species, and N is the number of all individuals of all species.

It was later converted to Simpson’s Index of Diversity, 1–D.

The average age of the stands was determined based on the age of a dozen trees from the dominant species using a Pressler (incremental) drill and data indicating the time of tree planting, which came from the city’s spatial development archives.

#### 2.3.2. Floristic Inventory

During the mapping of the plant communities in the field, a list of vascular plant species was developed for the entire study area with particular reference to the division into transects I, II and III. Transects I and II do not essentially differ from each other, so the information for them is given together. Plant species were determined by a key to vascular plant identification [57]. Plant names are given according to Flowering Plants and Pteridophytes of Poland [58].

#### 2.3.3. Ecological Requirements of Species

The research area is diversified in terms of habitat conditions and is conducive to species diversity in terms of habitat requirements. The gathered species both on and off transect plots were analyzed in terms of habitat conditions. This differentiation was analyzed using the ecological indicator values of vascular plants and their syntaxonomic affiliation according to Zarzycki et al. [59].

The flora collected were then analyzed in terms of ecological parameters such as plant life form, temperature (T), light (L), trophy (Tr), soil moisture (W), soil acidity (R) and organic content value (H). In this way, the habitat requirements of these flora were characterized by providing a numerical value (5-degree scale) of the given indicators. On the basis of these ecological indicators, the percentage share of flora for selected ecological parameters was calculated (Appendix A).

For the life histories according to population strategies (i.e., response to external stimuli), we used the classical Grimm division into C/S/R (competitive/stress/ruderal) strategies, and their combinations were also analyzed [60,61].

## 3. Results

The transformation of vegetation and the use of the Biała Przemsza valley is shown in Figure 3. The 1870 map shows the dominance of vegetation associated with swampy and wet habitats in the valley zone, and around it, in dry areas, there are pine forests. At that time, the river flowed along a natural riverbed and had many branches. On the map from 1930, it is not possible to determine the type of vegetation, and here, only meadows and forests are distinguished. During this period, areas of sand mining were established (Figure 3). On the right bank of the Biała Przemsza River, the loose development of the later settlement is marked (Figure 3).

The vegetation cover had changed little by 1960, and the decrease in vegetation coverage is associated with an increase in the area of development and railroad and mining infrastructure. During this period, the river flowed about 200 m from the boundary of the sand mine. The most serious environmental disturbance was the relocation and regulation (straightening) of the channel of the Biała Przemsza River in the 1970s (Figure 3) to the south east. This undoubtedly had a negative impact on the vegetation cover. The ecological system was completely disrupted, and the wetland vegetation changed to communities of indeterminate syntaxonomic rank. The relocation of the riverbed allowed for the expansion of the sand exploration area. Since the late 1980s, changes have occurred in the area, mainly in the urbanized zone. The current vegetation differs from the potential vegetation, resulting from changes in habitat conditions associated with engineering and reclamation work. The distribution of the current vegetation is shown in Figure 4. The sand mining pits are filled with coal mine waste [62]. After reclamation, the land is allocated for investment.

### 3.1. Forest Communities

In the zone of the regulated river and oxbow lakes, an artificial mixed forest was created as a result of reclamation. The vegetation distribution profile shown on the transects clearly indicates a regular distribution of tree populations (e.g., *Pinus sylvestris)*, which indicates the artificial introduction of species (Figure 5). A significant proportion of *Padus serotina* was found on transects I and II (Table 1). In the zone of the former riverbed and in the area of the depression, mainly deciduous species are found. In contrast, in the unbroken zone, no such distribution of species is observed, and riparian *Fraxino-Alnetum* can be found.

In the form of lines and scattered along the entire concrete channel as a result of succession, a community of assemblages of the class *Salicetea purpureae* is developing. For the most part, the riverbed slopes (embankments) are stabilized with *Spiraea salicifolia* (Figure 1D and Figure 5). This is a species that tolerates a wide range of pH values, substrate moisture and sunlight, and for this reason, it is often used for reclamation work.

**Artificial mixed forest**. On transect one (I) and transect two (II), a mixed forest community was found, consisting mainly of *Pinus sylvestris, Betula pendula, Quercus rubra* and *Robinia pseudacacia*, which form a layer of trees (Figure 5 and Table 1). The age of the tree stand is estimated at 40 years. The shrub layer is dominated by *Padus serotina* and *Frangula alnus*, which have a 25% degree of coverage in relevés 2 and 6 (Table 1). The forest undergrowth is diverse, which is mainly due to habitat and microhabitat conditions related to humidity and the degree of insolation of the area. In the zone of occurrence of the old riverbed, there are mainly species from the class *Molinio-Arrhenatheretea,* such as *Leontodon hispidus, Holcus lanatus, Prunella vulgaris* and others (Table 1). At the edge of the forest and in the gaps between trees, taxa characteristic of sandy areas of the *Koelerio glaucae-Corynephoretea canescentis* class develop, especially those of the order *Corynephoretalia canescentis*. Vascular species such as *Corynephorus canescens, Festuca ovina, Thymus pulegioides* and *Hieracium pilosella* make a significant contribution (Table 1). The bryophytes are represented mainly by *Polytrichum piliferum* and, to a lesser degree, *Ceratodon purpureus*.

In addition to diagnostic species for the syntaxons *Vaccinio-Piceetea* and *Dicrano-Pinon, Koelerio glaucae-Corynephoretea canescentis* and *Molinio-Arrhenatheretea*, significant numbers of high-coverage taxa are represented as accompanying and sporadic species (Table 1) belonging to various syntaxonomic groups (Appendix A). Simpson’s diversity index varies within the surveyed transect surfaces (Table 2), with the highest values of 0.8 and 0.82 obtained in its beginning sections near the old riverbed.

***Fraxino-Alnetum*** was analyzed on the surface (transect III) undisturbed by water engineering. This complex occurs in the form of a tree line along the river, and its width varies from 3 m (at the narrowest section) to 15 m (at the widest section). It has a multilayered vertical structure typical of natural riparian forests. The stand is mainly composed of *Alnus glutionosa,* accompanied by *Betula pendula, Fraxinus exelsior, Salix alba* and *S. fragilis*. This assemblage is characterized by an almost complete species composition, including species distinguished for the association, such as *Lycopus europaeus, Solanum dulcamara, Scutellaria galericulat* and *Lysimachia vulgaris* (Table 3). This also concerns species characteristic of the *Alno-Ulmion* union and the *Querco-Fagetea* class. All the reported species on the phytosociological relevé (Table 2) are associated with moist and wetland habitats (Appendix A) showing the normal water cycle in the phytocenosis and thus indicating the absence of any anthrogenic impact of humans.

This community is characterized by quite high diversity indicators (Table 4), ranging from 0.59 to 0.83.

**Non-forest communities** grow in regulated and unregulated sections, and they differ primarily in the size and nature of the surface. They are mainly represented by reeds from the *Magnocaricion* association and include *Caricetum paniculatae*, *Caricetum gracilis, Phalaridetum arundinaceae, Iridetum pseudacori* and species from the order of *Phragmitetalia* (*Phragmitetum australis, Glycerietum maximae*). In the anthropogenic sections, there are small patches of rushes with *Carex paniculata*, *Phalaris arundinaces* and *C. gracilis*, sometimes occurring as single clumps at various distances from each other (Figure 1A). In the section of the Biała Przemsza River unaffected by regulation works, the rushes are characterized by a typical species composition and structure related to the morphological variation of the riverbed, as is the case for natural systems. *Phragmites* and *Carex gracilis* occur universally. There are also areas with *Glyceria maxima* and *Iris pseudacorus*. Typically, the areas overgrown with *Phalaris arundinacea* and *Eupatorium cannabinum* are extended along the river in narrow belt forms. The largest patches are located in floodplains and depressions of land with limited water drainage and in the zone of beaver activity.

In contrast, the unregulated section does not cover a large area; hence, the number of communities is small. In the zone with water engineering, communities occur in small areas and are scattered along the river, and in transect III, their area is slightly larger, and they grow on mid-channel sandbars and in the bank zone (Figure 4).

In the industrial zone located in the northern part of the valley (in the place of the former riverbed) and covered with industrial and horticultural debris, there are ruderal species belonging to the class *Artemisitea vulgaris*, mainly represented by assemblages of *Artemisio-Tanacetetum vulgaris, Echio-Melilotetum* and *Dauco-Picridetum hieracioidis*. In small areas, there are also communities from the order of *Sisymbrietalia* and *Polygono-Chenopodietalia*. 

### 3.2. Floristic Diversity and Ecological Indicator Values 

A total of 150 species of vascular plants belonging to 52 botanical families and 114 genera have been identified (Appendix A). The best represented are the *Compositae* (17), *Poaceae* (13), *Caryophyllaceae* (11), *Rosaceae* (10) and *Apiaceae* (5) families (Table 1). 

In terms of syntaxonomic affiliation, most of the flora analyzed in transect I/II are represented by the classes *Vaccinio-Piceetea*, *Quercetea robori-petraeae*, *Festuco-Brometea*, *Molinio-Arrhenatheretea*, the order *Corynephorion canescentis*, *Arrhenatheretalia elatioris* and the *Dicrano-Pinion* union. On the other hand, transect III, passing through a relatively homogeneous habitat, is characterized mainly by riparian species (Table 2, Appendix A), and these belong mostly to the *Alnetea glutinosae* class and the *Alno-Ulmion* union, typical for wet areas along flowing rivers.

In terms of the life forms in all three transects, the biological spectrum is hemicryptophyte (Figure 6). In transect III, hydrophytes dominate, whereas in transect I/II, therophytes are more common. These differences result from the degree of succession development (transect I/II) and habitat conditions.

The categorization of species according to their light preferences (Figure 7A) shows the dominance of species adapted to half-shade conditions (18.27%-Transect-I/II; 35%-Transect-III) and moderate light (56.25%-Transect-I/II; 50%-Transect-III). On the other hand, transect I/II is dominated by species associated with full light (21.25%). As for the thermal preferences in the composition of the flora, the largest group comprises species adapted to moderately warm climatic conditions (Figure 7B). A small proportion of species (1.27%) have higher thermal preferences and occur only in transect I/II, and these are mainly psammophilous taxa (*Festuca psammophila, Koeleria glauca, Cirynephorus canescens*). In terms of humidity, species preferring moist (33.3%) and wet habitats (20.8%) dominate in transect III but not in transects I/II (Figure 8A).

The composition of the flora is also diverse due to the habitat requirements associated with the soil (pH) within the transects. Plants adapted to soils with a slightly acidic to neutral reaction prevail (Figure 8B), which also affects the richness of the substrate in nutrients (Figure 8C), which are mainly oligotrophic (Transect I/II) and mesotrophic (Transect III). Almost 90% of the flora from transect I/II develop on soils that are poor in humus and mineral-humus soils (Figure 8D). 

The flora that were analyzed are characterized by a variety of life strategies (plant strategies) and are mainly represented by competitive ruderals (CR), stress-tolerant species (S), stress-tolerant ruderals (SR), stress-tolerant competitors (SC) and species with mixed strategies (CR/CSR, S/SC, C/CSR) (Appendix A). These are typical strategies for ecological systems in the initial stage of development and under various degrees of anthropogenic stress. In contrast to transect III, on the surface of transect I/II (Appendix A), plants grow in conditions where stress is low and competition (C-R, S-R) is limited due to the existence of unoccupied ecological niches.

## 4. Discussion

In recent decades, rivers and valleys have been losing their naturalness primarily in urban and suburban areas associated with human economic activity and related adaptations. In the study area, this was most often associated with radical transformations as a result of regulation, including channel straightening, increasing slopes, unifying cross-section shapes and dimensions, eliminating bank and bottom irregularities, destroying ecotones along with ecological systems, cutting off connections between oxbow lakes and the main channel and reducing the extent and duration of valley floods [5,63]. In the case of the study area, this concerned changes in the riverbed, which was moved by 200 m to the south on average (Figure 1 and Figure 3). The key drivers of changes to the geographic space of the area were the need to enable underground coal mining and the protection of sand pits from flooding. The relocation of the Biała Przemsza riverbed was a necessary condition for the continuation of deep coal mining. The further exploitation of the sand without this procedure would have led to the drying up of the adjacent riparian ecosystems and the flooding of the excavation. These hydro-technical activities led to a drastic change in the water conditions that ensured the proper functioning of the wetland ecosystems developing in the area. Ecosystems related to the river valley functioned properly until 70 years ago (Figure 3). After the construction of the new riverbeds, they were gradually degraded due to water loss, and some of them were also mechanically damaged during hydro-technical works. The map of Poland’s potential vegetation shows that the area was overgrown with fertile deciduous forests such as *Tilio-Carpinetum, Dentario enneaphylli-Fagetum, Querco-Pinetum* on hills and *Fraxino-Alnetum* in wetlands and river valleys [63].

### 4.1. Diversity of Rush Communities

Vegetation on anthropogenic and semi-natural sections of the Biała Przemsza valley is difficult to compare with natural and semi-natural ecological systems from other regions of Poland or Europe, especially in the case of non-forest communities. Depending on the geomorphological conditions [64] and the location of the river valley (lowland or mountainous), the vegetation can vary significantly both in terms of species composition [65] and phytocenology [34,43,66].

Concrete escarpments and the colmatation of the bottom of the artificial riverbed prevented hydration of the adjacent area, limiting the growth of hygrophytes and hydrophytes. After 50 years, rushes have not developed in the coastal zone, because they cannot take root on the concrete plate. The bank vegetation is sparse and does not form dense patches; in places, it is even absent. Rushes from the Magnocaricon union only occur in places as a narrow belt on clay-silt sediments transported from the upper section of the river. There are also canals here that discharge municipal wastewater into the river, which affects the water quality and living organisms.

This may indicate that, despite the highly anthropogenic nature of the riverbed, the initial stages of regeneration of the phytocenosis typical of river valleys are observed. In this case, such developments can only be found in the zone along the riverbed. A similar situation was noticed in the plant communities covering the Brynica banks, which are relatively poor, as is the nearby bottom vegetation, especially in areas paved with chiseled plates and stones [34].

In the stretch of the Biała Przemsza not covered by regulation works, transformation processes of communities of rushes occur naturally, as in other areas without the influence of human activity [64,67]. A significant impact here comes from the elements of the riverbed relief and often the activity of beavers, which shape the riverine ecosystems (Figure 9).

### 4.2. Forest Vegetation Transformation

Artificial Scots pine forests and *Betula pendula* are developing in the transformed areas between the old and new riverbed created by the engineering work. They are species diverse in different parts of the site (Table 1) as a result of habitat conditions and varied topographic diversity (artificial embankment, bottom, slopes, old and new trough) associated with reclamation and engineering works. Similar ecological systems have been studied in other areas, and the authors [68] can confirm that all these communities, due to the dominance of Scots pine in the tree layer, can be conditionally included in the order *Pinetalia sylvestris* as non-hierarchical phytocenons. Often, *P. sylvestris* is accompanied in high cover by *Padus serotina* or *Molinia caerulea*. The considerable share of diagnostic species for the class *Vaccinio-Piceetea* and the association *Dicrano-Pinon* in the phytocenoses studied indicates the transformation of these systems towards pine forests of the *Leucobryo-Pinetum* type. Within the stand, gaps between canopies occur and facilitate high light availability for the forest floor (undergrowth). Many species of heliophytes (plants that thrive in bright light) characteristic of sand grasslands have been recorded in such places, mainly from the *Koelerio glaucae-Corynephoretea canescentis* [69]. Such diversity was reflected in Simpson’s diversity index. This diversity is due to the presence of a diverse anthropogenic relief created by regulatory work. This mainly refers to the presence of a large number of depressions and microdepressions that create a mosaic of habitats from moist to dry, allowing the entry of plant species with different ecological requirements.

A notable feature of these anthropogenic and artificial forest ecosystems is the presence of non-forest species (Table 1), which enjoy optimal conditions here and benefit from changes caused by forest management and other anthropogenic processes of a global nature [29,68,70,71,72].

Of the species considered highly invasive in Europe [73], *Quercus rubra*, *Robinia pseudoacacia* and *Prunus serotina* were found in the study area, fully meeting the definition of “transformers”, namely, species that change the structure of ecosystems [74]. Their high proportion in forest plantations poses a serious threat of transforming native forest communities in the immediate vicinity, as previously reported from many tropical [75] and subtropical [76] forests. They start to modify the structure of ecosystems significantly, e.g., by limiting the growth of seedlings [73], directly competing with native species and changing the physical, chemical and biotic properties of environment [73,74].

*Fraxino-Alnetum* develops in the form of a long and narrow belt (15 m on average) in the impact zone of spring flooding. The forest has a vertical multilayered structure that is typical of natural [68] and semi-natural riparian forest [77]. The geomorphological diversity of the riverbanks favors conditions for the development of species with different requirements and ecological adaptations [78]. Species composition and habitat condition in the area do not differ from similar ecological systems [5,33,79]. These ecosystems are not subject to the devastation caused by the regulation of the river channel or water melioration and are periodically waterlogged or flooded in fragments throughout the year compared to the artificial steep slopes on the regulated section, which has a crucial influence on the formation and functioning of wetland ecosystems.

Habitat conditions affect the floristic composition and structure of the plant ecosystems, and drainage causes a significant reduction in wetland biodiversity. Such human activities have caused the disappearance of wetland habitats in the Biała Przemsza area. Identifying the conditions of degraded habitats will allow plans to be drawn up for their restoration, during which efforts should be made to restore their natural conditions by regulating groundwater levels.

### 4.3. Flora Diversity and Ecological Parameters

The existence of relatively diverse flora in a small area is related to the presence of different habitat conditions and functional traits of plants. This is due to the predominance of anemochores in these families (*Poaceae, Compositae*), which produce large quantities of light seeds, allowing them to spread over considerable distances [57,80]. An important element indicating the stability of ecosystems is plant life strategies [60], which change under the intensity of stress (S), competition (C) and disturbance (R), as can be seen in the flora composition of the transects studied (Appendix A). Similarly, results were obtained in post-industrial areas with varying intensities of anthropopression [80].

The flora in the regulated (Transect I and II) and non-regulated (Transect III) sections were substantially different. Though in the case of the unregulated section, the flora is characterized by lower diversity in terms of the ecological number of indicators analyzed (T, L, W, R, Tr, H), in the case of the regulated section, there is increased diversity resulting (Figure 7 and Figure 8 and Appendix A), as mentioned above, from the presence of habitat mosaics. The high species diversity on anthropogenic surfaces results from many free ecological niches, varieties of habitats (from dry to wet), open areas (although not very large) and reclamation processes. Additionally, regarding the higher plant species variability in the reclaimed areas, this is characteristic of the area. It is probably due to the more nutrient-rich soil substrate created in the reclamation phase. This reclamation stage formed a layer of subsoil made up of humus, weathering, etc. that is thicker than the original soil profile on the fluvioglacial sands, as suggested by Rostański [80]. 

The results of the analysis of the spectrum of ecological indicators of the flora in the studied sections of the Biała Przemsza valley showed that its composition is dominated by anemochoric plants (e.g., *Compositae, Poaceae*) and heliophytes (Figure 7A and Appendix A), preferring moderate thermal, humid conditions and with a high diversity in terms of trophic requirements (Tr) and soil reaction (Figure 8). These species can tolerate a wide range of microhabitats and ecological factors according to results that have been obtained in similar areas [34,43,67,77,79].

### 4.4. Socioeconomic Aspects of Valley Use

Until the middle of the 20th century, the Biała Przemsza and its surroundings were a popular place of recreation (bathing in the river and sunbathing on its banks) for residents of nearby Sosnowiec, Jaworzno and neighboring towns. In 1930, a large resort, popularly known as “Skałka,” was built. A reminder of those days can be found in the names of some of the streets in Sosnowiec: Plażowa (Beach Street), Wczasowa (Summer Street) and Letniskowa (Summer Street). The deterioration of water quality since the mid-20th century, the channelization of the river and the creation of a spatial barrier (a large sand pit) prevented the residents of Sosnowiec from reaching the river, resulting in the cessation of swimming. However, as in many other places in the region [11], the riverside forests in the post-mining areas are important places for residents to walk. The river itself is used by kayakers.

The Biała Przemsza river valley and its surroundings boast recreational merits, and at the same time, there are no valuable habitats that could be the basis for legal protection. These areas are managed by cities and business entities not interested in protection for various reasons. Despite the processes of natural succession, which led in part to the formation of forest, this area is still treated as utilitarian, as a space for timber harvesting, a space for the development of new economic investments and a space for the management of water flow in the vicinity of railway lines [62,81,82].

## 5. Conclusions

Biocenotic systems are among the dynamically changing elements of the geosystem; hence, each of them will sooner or later be replaced by another. Changes in ecosystems are caused by various factors, both natural and anthropogenic, the effects of which (after about 50 years) were found in the valley of the Biała Przemsza River.

As a result of engineering work during the creation of the new channel, oxbow lakes were filled in, side arms were cut off, ponds were created in floodplains, a mosaic of wetlands was eliminated, and dry areas were created, which affected the diversity of flora. This effect on diversity is short-lived: after the formation of the climax communities, early-season species will be replaced by late-season species.After the riverbed had been moved to a new location, and the associated construction work had been completed, the vegetation cover associated with wet habitats such as wet meadows, marshes and thickets and riparian forests was completely destroyed.The formed and leveled land was reclaimed for forestry, and an artificial mixed forest is now developing. Despite its anthropogenic nature, it is species rich due to the presence of a habitat mosaic associated with the variability of the terrain (open sunny surface, water-filled depressions).The remarkable share of species diagnostic for the *Vaccinio-Piceetea* class and the *Dicrano-Pinon* association in the artificial phytocenoses of mixed forests indicates the transformation of these systems towards pine forests of the *Leucobryo-Pinetum* type.On a fragment of the unregulated Biała Przemsza River with natural relief, an alder riparian forest is developing. It is characterized by the presence of species diagnostic for this type of ecological system, indicating that despite its proximity to a regulated section, it is functioning properly.

## Figures and Tables

**Figure 1 ijerph-20-02255-f001:**
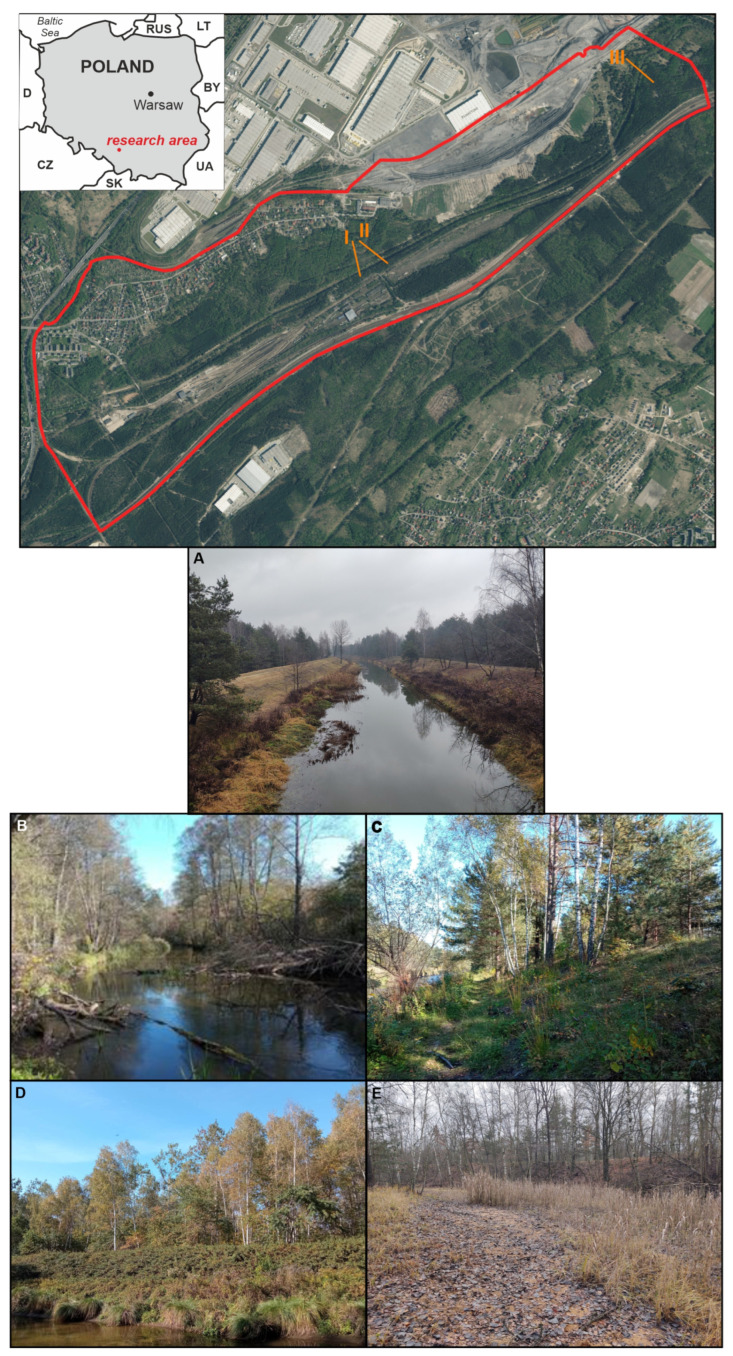
Location of study area: (**A**) general view of the straightened deposit together with a narrow patch of rushes; (**B**) unregulated section of the river with riparian forest; (**C**) mixed forest on the slope of an artificial embankment. (**D**) The slopes and tops of the embankment are fortified by *Spiraea salicifolia*, and the bank of the riverbed by a belt of rushes and (**E**) reed rushes on the old riverbed, where occasional water collects on the impermeable substrate.

**Figure 2 ijerph-20-02255-f002:**
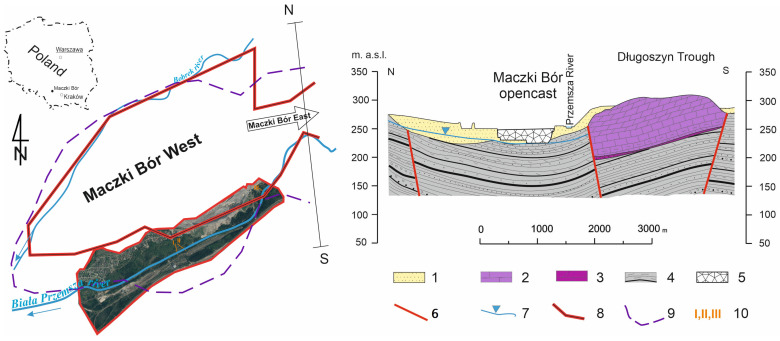
Location of the former Maczki Bór opencast and geological cross-section: 1—Pleistocene sands, 2—Middle Triassic, 3—Lover Triasic, 4—Upper Carboniferous with coal beds, 5—waste from coal mines, 6—fault, 7—groundwater level, 8—boundary of sand deposit, 9—boundary of the mining area, 10—transect numbers in the study area.

**Figure 3 ijerph-20-02255-f003:**
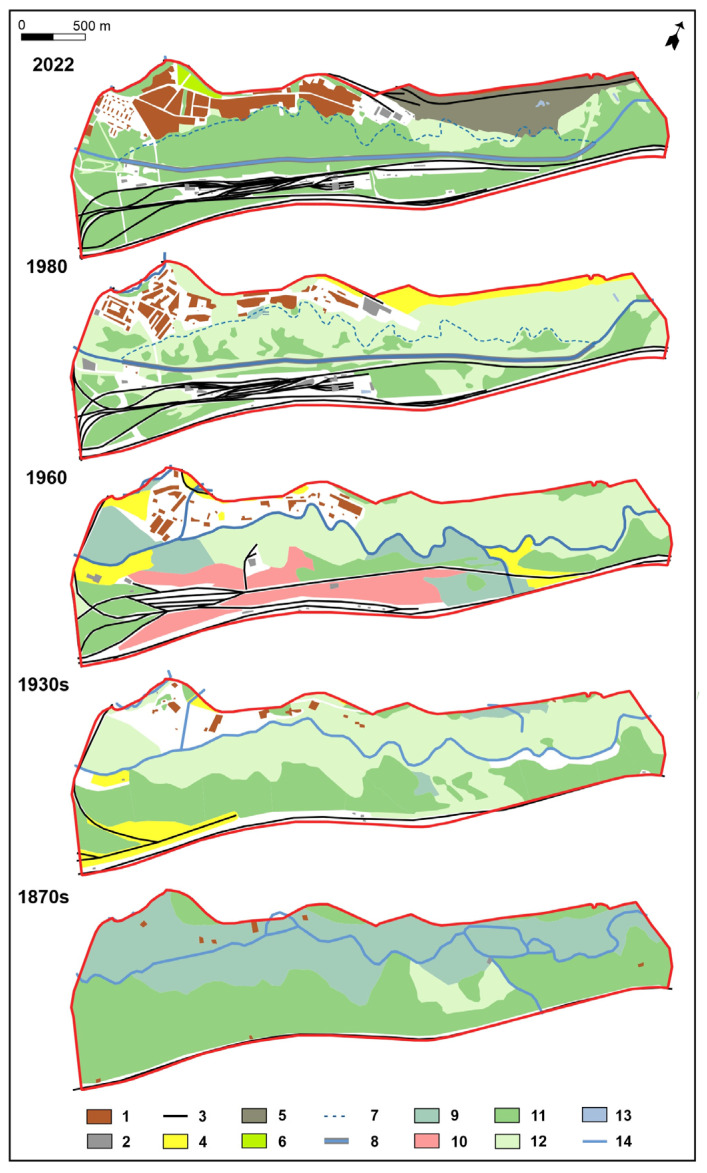
Land use/cover and vegetation changes in the period of 1870 to 2022: 1—buildings, 2—industrial buildings, 3—railway infrastructures, 4—sand and gravel quarry, 5—disposal of post-mining waste, 6—allotment gardens, 7—old riverbeds, 8—regulated riverbed, 9—marshy and moist meadow, 10—alder riparian forest, 11—forest, 12—meadow, 13—water reservoirs and 14—river.

**Figure 4 ijerph-20-02255-f004:**
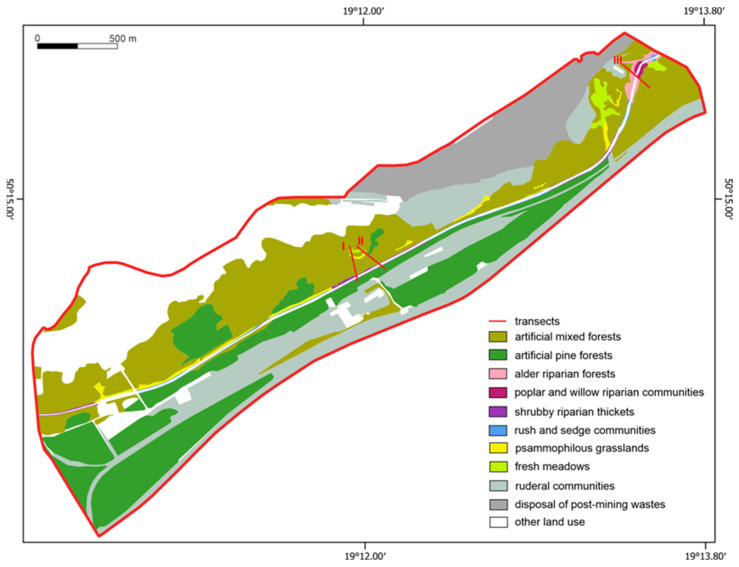
The distribution of modern contemporary vegetation communities in the study area.

**Figure 5 ijerph-20-02255-f005:**
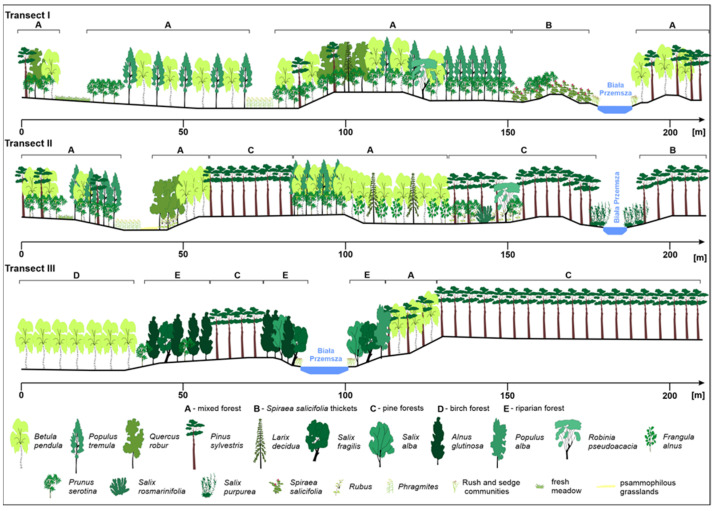
A scheme of the distribution of vegetation in the landscape profile starting from the old riverbed through various anthropogenic forms to the regulated riverbed or beyond.

**Figure 6 ijerph-20-02255-f006:**
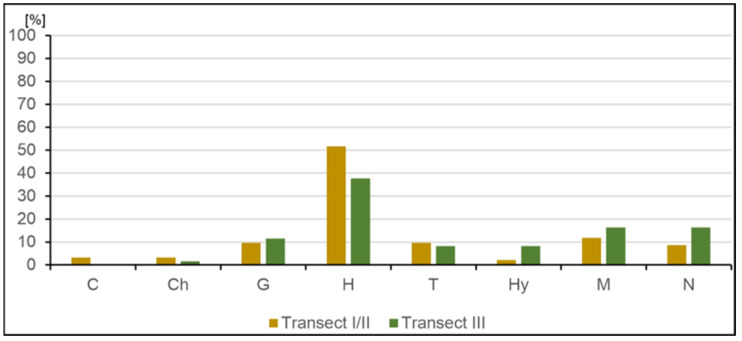
Percentage share of plant life forms: N—nanophanerophyte, M—megaphanerophyte, Hy—hydrophyte, T—therophyte, H—hemicryptophyte, G—geophyte, Ch—chamaephyte, C—herbaceous chamaephyte.

**Figure 7 ijerph-20-02255-f007:**
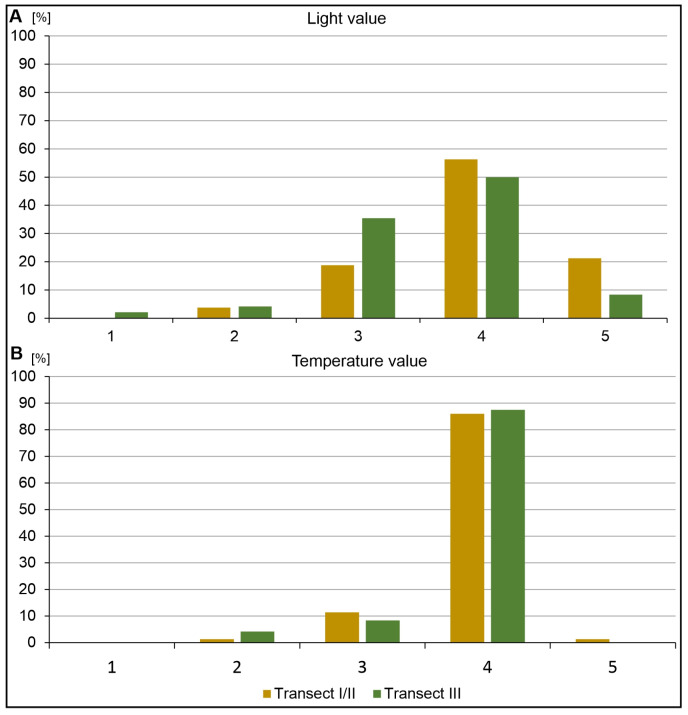
The differentiation of species according to light (**A**) and temperature (**B**) requirement value: A–1—deep shade; 2—moderate shade, 3—half shade, 4—moderate light, 5—full light; B–1—coldest regions in country; 2—moderately cold areas; 3—moderately cool climatic conditions, 4—moderately warm climatic conditions; 5—warmest regions and microhabitats.

**Figure 8 ijerph-20-02255-f008:**
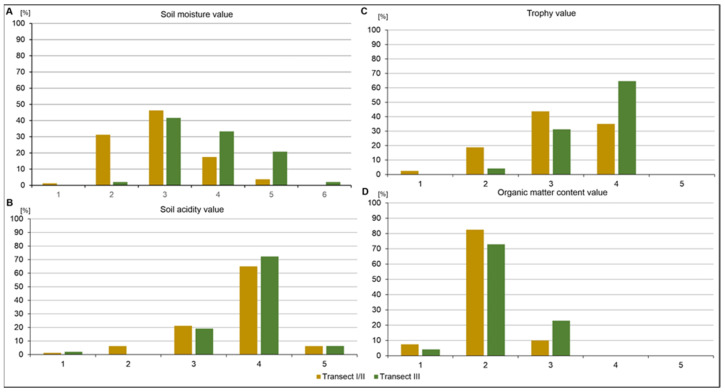
The differentiation of species according to soil moisture (**A**), soil reaction (**B**), trophy (**C**) and organic matter content (**D**) requirement values: A–1—very dry; 2—dry; 3—fresh; 4—moist; 5—wet; 6—aquatic; Bacidic soils, 4 ≤ pH < 5; 3—moderately acidic soils, 5 ≤ pH < 6; 4—neutral soils, 6 ≤ pH < 7; 5—alkaline soils pH ≥ 7; C–1—soil extremely poor (extremely oligotrophic); 2—soil poor (oligotrophic); 3—soil moderately poor (mesotrophic); 4—soil rich (eutrophic); 5—soil very rich (extremely fertile) and D–1—soil extremely poor (extremely oligotrophic); 2—soil poor (oligotrophic); 3—soil moderately poor (mesotrophic); 4—soil rich (eutrophic); 5—soil very rich (extremely fertile).

**Figure 9 ijerph-20-02255-f009:**
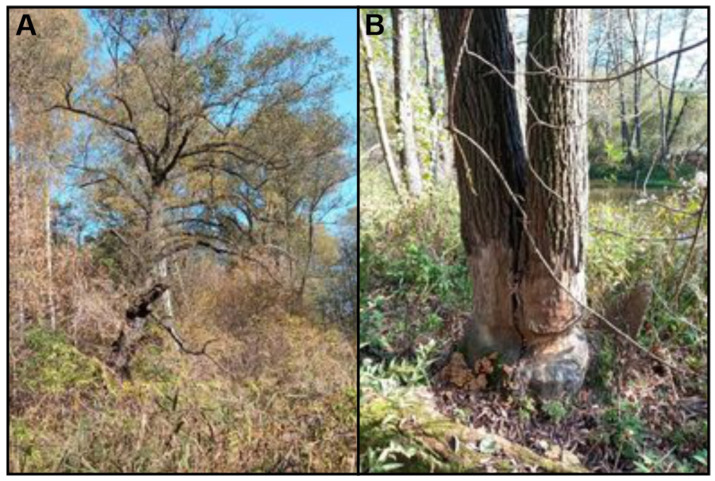
Fragment of the riparian forest (**A**) and traces of beaver activity (**B**).

**Table 1 ijerph-20-02255-t001:** The floristic composition of vegetation of the artificial mixed forest.

Number of Relevé	1	5	6	3	4	2	7	8	9
Location of Relevé	Transect I						Transect II		
Density of tree layer a [%]	60	80	90	-	90	30	80	90	80
Density of shrub layer b [%]	5	60	10	-	60	10	5	5	10
Cover of herb layer c [%]	30	10	10	90	-	70	10	5	5
Cover of mosses layer d [%]	5		-	-	-	-	-	-	-
Area of relevé in m^2^	100	100	100	100	100	100	100	100	100
Mean diameter of trees [cm]	20	28	22	-	15	-	30	15	20
Height of highest tree [m]	20	25	15	-	15	-	20	15	20
Number species in relevé	39	21	18	12	11	23	23	13	28
**I. Ch Cl. *Vaccinio-Piceetea + Dicrano-Pinon***									
*Pinus sylvestris* a	2	2	5	.	+	.	4	5	4
*Pinus sylvestris* b	+	+	+	.	.	.	1	+	+
*Pinus sylvestris* c	+	+	+	.	.		+	+	+
*Pyrola minor* c	+	+	.	.	.	.	.	.	+
*Pyrola rotundifolia* c	+	+	.	.	.	.	+	.	+
*Vaccinium vitis-idaea* c	+	+	+	.	.	.	+		+
*Deschampsia flexuosa* c	+	.	+	.	.	.	+	.	+
*Chimaphila umbellata* c	.	+	.	.	.	.	.	.	+
**II. Ch Cl. *Koelerio glaucae-Corynephoretea canescentis***									
*Armeria elongata* c	1	+	.	.	.	.	+	+	.
*Cardaminopsis arenosa* c	+	.	.	1	.	+	.	+	+
*Corynephorus canescens* c	1	+	.	+	.	+	+	.	.
*Festuca ovina* c	+	1	.	.	.	.	1	.	+
*Jasione montana* c	+	.	.	.	.	.	+	+	+
*Hieracium pilosella* c	+	.	.	.	.	.	1	+	+
*Rumex acetosella* c	+	+	.	.	.	.	+	.	+
*Thymus pulegioides* c	2	.	.	.	.	.	+	.	.
*Polytrichum piliferum* d	+	.	.	.	.	.	1	.	.
**III. Ch. Cl. *Molinio-Arrhenatheretea***									
*Leontodon hispidus* c	+	.	+	.	.	.	.	.	.
*Holcus lanatus* c	+	.	+	.	.	.	.	.	+
*Plantago lanceolata* c	+	.	.	.	.	.	.	.	.
*Prunella vulgaris* c	+	.	.	.	.	.	.	.	.
*Euphrasia rostkoviana* c	+	+	+	.	.	.	+	.	.
*Achillea millefolium* c	+	.	.	.	.	+	.	.	+
*Molinia caerulea* c	.	+	.	.	.	1	+	.	+
**IV. Accompanying species**									
*Acer platanoides* a	+	.	.	.	.	.	.	.	.
*Acer pseudoplatanus* a	+	.	.	.	+	.	.	.	.
*Betula pendula* a	3	3	1	.	.	1	+	1	1
*Quercus robur* a	.	+	+	.	.	+	.	.	+
*Quercus rubra* a	.	1	+	+	.	.	.	.	1
*Populus tremula* a	.	+	+	+	5	.	.	+	.
*Populus nigra* a	1	1	.	.	.	.	.	.	.
*Robinia pseudacacia* a	.	1	.	.	+	.	.	+	+
*Frangula alnus* b	.	+	2	.	.	2	.	+	1
*Padus serotina* a	.	.	.	.	.	1	.	.	+
*Padus serotina* b	1	3	1	.	3	+	.	1	1
*Viburnum opulus* b	.	.	.	.	+	+	.	.	.
*Agrostit canina* c	3	.	.	2	.	3	.	.	+
*Carex hirta* c	2	.	.	2	.	.	+	+	.+
*Calamagrostis epigejos* c	1	+	+	1	.	+	.	+	+
*Phragmites australis* c	.	.	.	4	.	1	.	.	.
*Potentilla argentea* c	1	.	.	.	.	.	+	.	.
*Rubus fruticosus* c	1	.	.	.	.	.	.	.	+
**Sporadic species:***Agrostis stolonifera* 1c(+); *Arenaria serpyllifolia* 1c(+); *Viola tricolor* 1c(+); *Scabiosa ochroleuca* 1c(+), 9(+); *Silene vulgaris* 1c(+); *Galium mollugo* 1c(+), 6 (+); *Rumex crispus* 1c(+); *Geranium pratense* 1c(+); *Cirsium arvense* 1c(+); *Lysimachia vulgaris* 2c(1), 3c(+), 9(+); *Cirsium oleraceum* 2c(1), 3c(+); *Aegopodium podagraria* 2c(1), 3(1); *Eupatorium cannabinum* 2c(+), 3(1); *Agrimonia eupatoria* 2c(+); *Chaerophyllum aromaticum* 2c(+); *Equisetum arvense* 2c(+); *Galeopsis tetrahit* 2c(+); *Humulus lupulus* 2c(+), 4(+); *Iris pseudacorus* 2c(+), 3(1); *Peucedanum oreoselinum* 2c(+), *Heracleum sphondylium* 3c(+); *Salix caprea* 4b(1); *Dryopteris filix-mas* 4c(+); *Sorbus aucuparia* b 4(+), 6(+), 9(+); *Ligustrum vulgare* b 4(+); *Hieracium lachenalii* 5c (+); *Salix rosmarinifolia* 6b(1), 9 (+); *Erigeron canadensis* 6c(+); *Echium vulgare* 6c(+); *Lupinus polyphyllus* 6c(+); *Salix acutifoilia* 7b(+); *Dianthus deltoides* 7c(+); *Hypericum maculatum* 7c (+); Sedum acre 7,8c(+); Erigeron acris 7c(+); *Rhinanthus minor* 7c(+);*Cytisus scoparius* 7c,9 (+); *Cornus sanguinea* 9b(+); *Daucus carota* 9c(+); *Hypericum perforatum* 9c(+).									

**Explanation:** the Braun-Blanquet [55] scale was used, in which the following categories were applied: degree 5—species covering 75–100% of the area; 4—species covering 50–75%; 3—species covering 25–50%; 2–25–5%, 1–5–1% and +-species is present in the area.

**Table 2 ijerph-20-02255-t002:** Simpson’s Index of Diversity in artificial mixed forest.

Number of Relevé	1	5	6	3	4	2	7	8	9
**Simpson’s Index of Diversity: 1–D**	0.86	0.76	0.46	0.7	0.48	0.8	0.45	0.21	0.46

**Table 3 ijerph-20-02255-t003:** The floristic composition of vegetation of the *Fraxino-Alnetum*.

Number of Relevé	1	4	3	5	2
Location of Relevé	Transect III				
Density of tree layer a [%]	80	90	80	80	60
Density of shrub layer b [%]	40	10	10	15	20
Cover of herb layer c [%]	30	60	50	40	30
Cover of mosses layer d [%]	-	-	-		-
Area of *relevé in m^2^*	200	200	200	200	200
Mean diameter of trees [cm]	40	60	70-	30	30
Height of highest tree [m]	20	15	20	25	25
Number species in *relevé*	31	19	22	21	27
**I. Ch. DAss. *Faraxino-Alnetum***					
*Frangula alnus* b	1	+	1	+	+
*Lycopus europaeus* c	1	.	+	.	.
*Solanum dulcama* c	+	+	.	.	+
*Scutellaria galericulata*	+	+	.	.	.
*Iris pseudacorus* c	+	.	.	3	1
*Galium palustre*	.	.	+	.	+
*Lysimachia vulgaris* c	+	.	1	.	1
**II. ChAll. *Alno-Ulmion + ChCl. Querco-Fagetea***					
*Alnus glutinosa* a	4	4	4	3	3
*Alnus glutinosa* b	+	+	+	1	1
*Fraxinus exelsior* a	+	+	+	1	+
*Padus avium* b	+	.	+	.	.
*Aegopodium podagraria* c	1	.	+	1	+
*Carex elongata* c	2	1	+	+	2
*Scirpus sylvaticus* c	3	+	1	.	1
*Humulus lupulus* c	2	2	+	+	+
*Phragmites australis*	.	.	+	.	1
**III. Ch. *Salicetea purpurea* + *Salicion albae***					
*Salix alba* a	+	1	+	+	2
*Salix fragilis* a	1	+	1	+	+
*Phalaris arundinacea* c	+	+	.	.	.
**IV. Accompanying species**					
*Betula pendula* a	3	.	.	3	1
*Populus tremula* a	.	5	+	+	.
*Pinus sylvestris* a	.	.	.	1	+
*Padus serotina* b	1	.	.	1	+
*Sambucus nigra* b	1	.	+	1	+
*Viburnum opulus* b	+	+	+	+	+
*Eupatorium cannabinum* c	1	1	+	+	1
*Cirsium oleraceum* c	1	.	1	.	+
*Chaerophyllum aromaticum* c	+	.	1	+	+
*Carex gracilis* c	1	1	.	2	+
*Galeopsis tetrahit*	.	+	+	+	.
*Rubus fruticosus* c	1	+	1	+	+
**Sporadic species:** *Rhamnus cathartica* 1b (+); *Valeriana officinalis* 1c (+); *Urtica dioica* 1c (+), *Agrostis stolonifera* 2c (+); *Galium aparine* 1c (+); *Geranium robertianum* 2c (+); *Impatiens noli-tangere* 1c (+); *Rubus caesius* 1c (+); *Equisetum palustre* 2c (+); *Salix rosmarinifolia* 4b (1), *Salix caprea* 4b (+), *Stellaria nemorum* 5c (+)					

**Table 4 ijerph-20-02255-t004:** Simpson’s Index of Diversity in *Fraxino-Alnetum*.

Number of Relevé	1	4	3	5	2
Simpson’s Index of Diversity: 1–D	0.78	0.68	0.59	0.83	0.82

## Data Availability

On reasonable request, all data can be received from the corresponding author.

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
