# Peer review of "Long-Term Vegetation Changes and Socioeconomic Effects of River Engineering in Industrialized Areas (Southern Poland)"

_ijerph, 2023, doi:10.3390/ijerph20032255_

Round 1

Reviewer 1 Report

The manuscript covers an interesting subject, the effects of anthropogenic changes on plant communities. The results of the manuscript are rather descriptive in nature.

(1) Why are transects I and II so close together? Would it not be more interesting to space them further apart or look at different anthropogenically influenced habitats along different transects?

(2) The discussion contains too many repetitions of results and not enough discussion in the context of the state-of-the-art literature. Why is the species diversity higher in anthropogenically influenced compared to natural transects? How does this compare to other studies, climatic zones and ecoregions? Overall, please discuss you results further in the context of the relevant literature and move some of the result descriptions that are currently contained in the discussion section to the results section (and merge them with the text in the results section).

(3) The English of the manuscript is ok, but the manuscript contains several out of place and difficult to understand expressions. I would suggest to have the whole manuscript proofread and corrected by a professional proofreading service.

I have provided more specific comments in the attached pdf file

Reviewer 2 Report

The review of manuscript ID: ijerph-2121855

The reviewed manuscript is analyzing plant biocenoses in anthropogenically transformed landscape of Biała Przemsza River valley (Poland). This area was transformed mainly by mining activities, which led to serious changes of ecosystems, especially natural wetland areas, belonging to natural riverine ecosystem. Biodiversity analysis is good tool for evaluation of existing ecosystems, giving information about ecosystem functioning and health.

The manuscript is interesting, and rather well written.

I have one major question. Authors gives in manuscript long-term changes of phytocenoses. This floristic analysis takes into consideration changes induced by direct anthropogenic activity in the area. But we live now in the time of global climate changes. Hence it is possible, that changes in phytocenoses are caused by two factors: direct human activity in the research area as well as the long -term climate changes. Could Authors take this issue into consideration in discussion?

Minor comments:

Lines 110-116 – this text from template should be removed;

Line 229 – insert “al” in the phrase “Zarzycki et. ;

Line 292 – Table 2 should be numbered as Table 1;

Table 3 – last line on page 11 – add “s” in the word “sylvestri”

All plant species in tables should be in expanded form eg. “Pinus sylvestris L.”;

In the References – all native language positions titles should be translated into English (in brackets).

Reviewer 3 Report

Comments for authors:

This study determined and compared the diversity and distribution of vegetation in the Bia.a Przemsza valley in sections of channel straightening with the old riverbed and undisturbed by engineering works against the background of land use in temporal and spatial aspects. This paper deals with relevant and interesting topic and is potentially a valuable contribution to large-scale biodiversity conservation. However, there are numerous problems, some of them major, that need to be addressed. The following is my primary concerns.

1. Please check format, typo, etc. There are numerous minor mistakes.

2. The title is vague to understand what this study has done. I recommend that the authors write what kind of organism they targeted in the title.

3. There are no hypotheses and predictions we can follow in this manuscript. I recommend that the authors describe the hypotheses and predictions on this study in the introduction of this paper.

4. There are descriptions (Line 110-116) that are not related to this study in the first paragraph in the materials and methods section.

5. The authors need to provide more detailed sampling design.

6. I could not find statistical analyses (except for descriptive statistic like bar charts) in this manuscript. I recommend that the authors use statistical analyses to examine relationships between the diversity and environmental factors.

7. The discussion seems to be out of focus. The authors need to better link their study objectives to the content of the discussion.

8. The conservation/management suggestions are limited in this study. I recommend that the authors write the more specific conservation/management strategies based on your results.

Round 2

Reviewer 1 Report

The authors have not sufficiently revised the manuscript. Especially the newly revised sections contain grammatical errors with English of unacceptable quality. I am happy to review the scientific content of a properly revised version of the manuscript that is clearly written and in grammatically correct English.

Reviewer 3 Report

I think that the authors appropriately respond to my comments. I recommend that the authors check the manuscript again to make sure if there are minor mistakes in the manuscript. Thank you for giving me the opportunity for reviewing this manuscript.

Author Response

The text was proofread by Robert Pagett, a native speaker from Poznan (School of Native English). We have also enclosed a certificate attesting to the detailed linguistic correction. The text has also been checked from the technical point of view. We hope that the text is now free of any linguistic flaws.

Round 3

Reviewer 1 Report

I have some minor suggestions for revisions outlined the attached pdf, otherwise the manuscript is acceptable for publication

Author Response

Thank you for your comments. Corrections were included in the manuscript.